# Effect of bis-TEMPO Sebacate on Mechanical Properties and Oxidative Resistance of Peroxide-Crosslinked Polyolefin Compositions

**DOI:** 10.3390/polym17243325

**Published:** 2025-12-17

**Authors:** Artem Chizhov, Aleksandr Goriaev, Svetlana Belus, Maksim Svistunov

**Affiliations:** 1Institute of Theoretical and Applied Electrodynamics, Russian Academy of Sciences, Moscow 125412, Russia; 2Institute of Fundamental and Applied Chemistry, P.G. Demidov Yaroslavl State University, Yaroslavl 150003, Russia; goryaes@gmail.com; 3LLC “Vestplast”, Pereslavl Zalessky 152025, Russia; log@vestplast.com; 4National Research Center “Kurchatov Institute”, Moscow 125182, Russia; sbielus@gmail.com

**Keywords:** polyolefins, TEMPO, bis-TEMPO, peroxide crosslinking, scorching, scorch retardants

## Abstract

TEMPO derivatives are well known as scorch retardants due to their ability to effectively quench free alkyl radicals during peroxide crosslinking of polymer compositions. However, in practice this leads to the loss of crosslinking density due to a irreversible decrease in the number of alkyl radicals involved in the crosslinking process. One approach to solving this problem is the use of TEMPO-based biradical molecules, which, on the one hand, are able to effectively quench alkyl radicals, and on the other hand, can couple macroradicals, partially compensating for the loss of crosslinking density. The aim of this work was to reveal the effect of bis(1-oxyl-2,2,6,6-tetramethylpiperidin-4-yl) sebacate (bis-TEMPO) in the concentration range of 0.11–0.44 phr on the delay in the onset of dynamic crosslinking of polyolefin composites initiated by peroxide, as well as the oxidative stability of the resulting crosslinked composites. The obtained data show that using bis-TEMPO at a concentration of less than 0.27 phr increases the crosslinking density of the polyolefin composite, with a crosslinking onset delay of up to 36 s achieved. Simultaneously, antioxidant functionality of bis-TEMPO in crosslinked composites is considered moderate and leads to an increase in the OIT values by 1.7–2.8 times. The crosslinking onset delay time under dynamic conditions is well described by a first-order kinetic model at a constant temperature. The obtained data confirm the efficiency and predictability of bis-TEMPO as a scorch retardant for polyolefin composites.

## 1. Introduction

Polymeric materials based on polyolefins are often essential for various applications, such as wire and cable insulation, pipes, and various construction materials [1]. One of the common methods of postprocessing polymers, which improves their temperature and chemical resistance, impact strength, wear resistance is crosslinking [2,3], consists of forming a three-dimensional network of bonds between linear or slightly branched polymer molecules. Various methods of crosslinking polyolefins are known, which are divided into physical methods (UV crosslinking, γ-radiation crosslinking, e-beam crosslinking) and chemical methods (peroxide crosslinking, silane crosslinking, azo crosslinking) [4]. Peroxide crosslinking has the advantage of being easy to implement and involves the use of common components, such as organic peroxides, as crosslinking initiators. However, various difficulties are known in the process of peroxide crosslinking. Insufficient mixing efficiency in combination with the high rate of peroxide decomposition can lead to non-uniform crosslinking, due to the fact that individual areas in the polymer melt will contain a higher concentration of peroxide, which will lead to excessive crosslinking and chain scission processes. This problem is known as “scorching” and can be solved by introducing special additives into the polymer composition that can trap alkyl radicals in the earlier stages, allowing delaying the onset of crosslinking and achieving a preliminary more thorough distribution of peroxide in the polymer blend, and as a result obtaining a crosslinked material with more uniform properties.

Despite the fact that many known molecules are capable of quenching radicals, the choice of an effective scorching retardant is a complex scientific and technological task [5]. It should be noted that the usual trapping of radicals by typical antioxidant molecules (hindered phenols, for example) leads to their irreversible removal from the crosslinking reaction, which inevitably leads to a loss of the density of the three-dimensional network; i.e., the delay in crosslinking is achieved at the cost of reducing the crosslinking density. To address this issue, an initial study suggested employing a α-methylstyrene dimer (MSD), which led to a delay in the onset of crosslinking as well as an enhancement in the crosslinking density. In the first step, MSD adds to macroradicals at the double bond, forming an intermediate adduct radical, which fragments to form a cumyl radical, while at the same time, the unsaturated fragment of the MSD molecule remains grafted to the polymer chain. In the second step, these unsaturated MSD residues react with macroradicals, producing a crosslinked polymer [6].

One of the promising types of scorch retardants are 2,2,6,6-tetramethylpiperidine-1-oxyl (TEMPO) derivatives [7,8,9]. The TEMPO molecule is a stable radical due to steric hindrance, capable of effective (rate constant $10^{8}$–$10^{9}$ $M^{-1}s^{-1}$ [10]) and selective trapping of alkyl radicals to form the corresponding N-alkoxyamines [11], which are stable at typical crosslinking temperatures. However, the use of the monoradical TEMPO molecule still carries the risk of loss of crosslink density because of the irreversible trapping of alkyl radicals. In a number of articles, various derivatives of TEMPO were developed, providing additional functionality to compensate for loss of crosslink density [12]. For example, in a series of articles [13,14,15], the authors gave preference to 4-acryloyloxy-TEMPO to delay the onset of the polyolefine crosslinking time without loss of the ultimate crosslink density in the temperature range of 140–180 °C. While the TEMPO part of the molecule is involved in the quenching of macroradicals at early stages of crosslinking, the grafted acryloxy groups can undergo oligomerization at later stages, increasing the crosslinking density. Compared with TEMPO derivatives containing different polymerizable groups (methacryloyloxy-, cinnamoyloxy-, crotonoyloxy-, and others) at the same position, 4-acryloyloxy-TEMPO showed the best efficiency due to the higher reactivity of the acrylic groups.

Another methodology consists of the use of biradical molecules such as bis(1-oxyl-2,2,6,6-tetramethylpiperidine-4-yl)sebacate (bis-TEMPO), which can demonstrate dual action: on the one hand, the nitroxyl groups can quench alkyl radicals, and on the other hand, they can form crosslinks between polymer chains by reacting with macroradicals and thus increase the crosslink density. However, there are conflicting reports in the literature regarding the feasibility of using bis-TEMPO as a retardant for maintaining the crosslink density. According to [16], the use of bis-TEMPO leads to an essential loss of crosslinking density, since the trapping of methyl radicals by the bis-TEMPO molecule is also a highly probable reaction that does not directly lead to additional crosslinking and, moreover, irreversibly reduces the concentration of radicals initiating crosslinking process. In contrast, the authors in [7] revealed an increase in the content of the gel fraction of crosslinked polyethylene at a temperature of 182 °C from 78% to 84% with the addition of 0.25 wt.% bis-TEMPO with a corresponding increase in torque during crosslinking. Some patents, such as [17], have also suggested the application of bis-TEMPO for its properties as a scorch retardant. However, the cited papers investigated crosslinking under static conditions using rheometers on premixed formulations, while studies on the effect of bis-TEMPO on crosslinking under dynamic conditions have not yet been published. Furthermore, the literature rarely covers the issue of the combined effect of TEMPO derivatives on the delay in the onset of crosslinking and the oxidative stability of the resulting crosslinked polymers.

This article presents studies of the effect of bis-TEMPO in a concentration of 0.11–0.44 parts by weight on the delay in the onset of dynamic crosslinking of polymer blends based on linear low-density polyethylene (LLDPE) and polyolefine elastomer (POE). This composition was selected because it provides, after crosslinking, suitable mechanical properties for unfilled insulation of some types of electrical cables. The crosslinking was carried out at a temperature of 180–210 °C using 2,5-dimethyl-2,5-di(tert-butylperoxy) hexane as an initiator and triallylisocyanurate (TAIC) as a crosslinking agent. The laboratory route of bis-TEMPO synthesis and confirmation of its molecular structure by ^1^H and ^13^C NMR are described in detail. The mechanical properties of the obtained crosslinked materials, oxidation resistance, and gel fraction content in them are discussed and compared.

## 2. Materials and Methods

### 2.1. Materials

4-hydroxy-2,2,6,6-tetramethylpiperidin-1-oxyl (4-hydroxy-TEMPO, 98%), sebacic acid (99%), and triallyl isocyanurate (TAIC) (98%) were purchased from Macklin (Shanghai, China). Benzoyl chloride, 99.8%, was purchased from Chemcraft (Kaliningrad, Russia). Thionyl chloride, 98%; triethylamine, 98%; HCl, 38%; Na_2_SO_4_, 99.9%, anhydrous; toluene, ≥98%; and hexane≥98%, were purchased from Vekton (Saint Petersburg, Russia). 2,5-dimethyl-2,5-di(tert-butylperoxy)hexane, ≥92%, (DBPH), Trigonox^®^ 101; tris(2,4-di-tert-butylphenyl)phosphite, ≥99%, Sonox^®^ 168; and pentaerythritol tetrakis(3,5-di-tert-butyl-4-hydroxyhydrocinnamate, Irganox^®^ 1010 FF, ≥99%, were also used.

### 2.2. Synthesis of bis-TEMPO

The bis-TEMPO synthesis route was optimized based on methods in the known literature [16,18,19].

#### 2.2.1. Sebacoyl Chloride

A sebacic acid weighing 110 g (ν = 0.545 mol) was combined with 100 mL of SOCl_2_ (ν = 1.378 mol). The reaction mixture was heated in a 1 L conical flask for 6 h to 50 °C under reflux while stirring with a magnetic stirrer. The gases released were bubbled through a saturated solution of Na_2_CO_3_ for neutralization. During the reaction, sebacoyl chloride was formed as a yellowish liquid at the bottom of the flask. After completion of the reaction, excess SOCl_2_ was removed on a rotary evaporator at 70 °C. The yield is quantitative.

#### 2.2.2. bis-TEMPO Sebacate

The synthesis was carried out in a 2 L double-walled glass reactor equipped with an overhead stirrer and connected to a circulation thermostat. 4-hydroxy-TEMPO (207 g, ν = 1.2 mol) was dissolved in 1500 mL of toluene; then 166 mL of triethylamine (ρ = 1.13 $g/{\mathrm{cm}}^{3}$, ν = 1.2 mol) was added. Synthesized sebacoyl chloride (124 mL) was added dropwise at 50 °C over 15 min. Subsequently, the reaction mixture was maintained at 40 °C for 2 h; following this, the heating was discontinued, and the reaction was allowed to proceed at room temperature (RT) for an additional 18 h. Each 1 L portion of the reaction mixture was then transferred into a 2 L separatory funnel and subjected to four washes with 0.5 L of a dilute HCl solution (10 mL HCl per 1 L of water). The organic phases were collected and dried over Na_2_SO_4_, and the solvent was removed using a rotary evaporator at 70 °C, yielding a viscous dark red liquid. Finally, 300 mL of ethanol (95%) was added to this liquid, and the resulting solution was combined with 300 mL of water at RT. The obtained two-phase liquid mixture was cooled to 0 °C, stirring periodically, resulting in the formation of orange crystals. If crystal formation was hampered, part of the aqueous phase (approx. 50 mL) was drained and replaced with ethanol. Bis-TEMPO crystals obtained were separated in a Buchner funnel, washed with the water–ethanol solution (1:1) and dried under vacuum at 60 °C for 12 h. The yield was 242 g (87% based on the starting sebacic acid). Before use, bis-TEMPO was recrystallized from hexane. We found C—66.07%; H—9.98%; N—5.23%. Calc: C—65.83%; H—9.80%; and N—5.4%.

#### 2.2.3. 4-Benzoyloxy-TEMPO

The synthesis and purification of 4-benzoyloxy-TEMPO (BzO-TEMPO) were carried out by similar procedures using 4-hydroxy-TEMPO and benzoyl chloride as starting materials.

### 2.3. Preparing of Polymer Samples

A mixture of industrial polymers was used for crosslinking, comprising linear low-density polyethylene (LLDPE), Lucene™ HP3518CN (1-hexene co-monomer), and polyolefine elastomer (POE), Lucene™ LC670 (ethylene-1-octene co-polymer), without additional purification in a weight ratio of 30:70. A standard package of antioxidants (hindered phenol and phosphite) and a crosslinking agent (TAIC) were also added to the polymer mixture. The studied compositions in phr are given in Table 1. The polymer granules were pre-mixed with each other and then loaded into the mixer chamber, followed by antioxidants, TAIC, bis-TEMPO, and finally peroxide. The total mass of the polymer mixture was 38 g. The compositions were then dynamically crosslinked using a Brabender measuring mixer (Anton Paar, Graz, Austria) for 15 min at 60 rpm and an initial temperature of 180 °C. Then, the resulting polymer mass was removed from the mixer and pressed at 175 °C and 6.86 MPa for 120 s to produce sheets 100×100 mm in size and 1 mm thick. Specimens for tensile measurements, DMA, gel fraction, and DSC measurements were cut out from these sheets.

Recording and processing of plastograms were performed using Brabender WINMIX software (v. 4.9.7).

### 2.4. Methods

NMR ^1^H and ^13^C spectra were recorded on a AVANCE III NanoBay (Bruker, Billerica, MA, USA) spectrometer in CDCl_3_ at 300 and 75 MHz, respectively. Since the bis-TEMPO molecule is a stable biradical, this leads to the ultimate broadering of signals from the protons of the piperidine ring. To obtain correct NMR spectra, 1–2 drops of phenylhydrazine were added to the NMR tube with the bis-TEMPO solution for selective in situ reduction of R−NO^•^ to the corresponding R−NH−OH groups [20,21].

The elemental composition of bis-TEMPO was determined using CHNS EuroEA 3000 elemental analyzer (EuroVector, Pavia, Italy).

Gel content in crosslinked samples was determined in accordance with the recommendations in [22] using a Soxhlet extractor and *o*-xylene as a solvent. Samples weighing 0.4–0.6 g were placed in a stainless steel mesh bag (0.125 × 0.125 mm cell) and extraction was performed for 22 h, with an average cycle time of 4–5 min. After extraction, the samples were air-dried for 12 h at 60 °C.

The determination of the oxidation induction time (OIT) and oxidation onset temperature (OOT) was carried out using a differential scanning calorimeter DSC200L, temperature measurement error <3 °C (Xiang Yi Instruments, Xiangtan, China). In the first case, the samples were initially heated in a N_2_ flow to 200 °C, after which the gas was replaced with pure O_2_ and the sample decomposition time was recorded. In the second case, the process was carried out in an air flow with continuous heating of the sample at 5 °C/min. OIT and OOT values were calculated using Termal Analysis software (v. 24.6.194).

Dynamic mechanical analysis was performed on a DMA1 system (Mettler-Toledo, LLC, Columbus, OH, USA) in tension mode at constant frequency of 1 Hz in the temperature range of 25–90 °C. The measurement of the ultimate tensile strength and elongation at break was carried out on a extensometer IR-5040-5 (“TOCHPRIBOR”, St. Petersburg, Russia) at a tensile speed of 100 mm/min.

## 3. Results

### 3.1. Characterization of bis-TEMPO

The study of the chemical structure of the synthesized bis-TEMPO was based on NMR spectroscopy of its product reduced with phenylhydrazine. The ^1^H NMR spectra obtained confirm the proposed structure of the synthesized compound (Figure 1a). Spectra are normalized to the residual peak of the solvent (7.26 ppm). Multiplets at 7.05–7.20 ppm and 6.68–6.80 ppm are related to the chemical shift of protons in the aromatic structures of phenylhydrazine and its oxidation products (e.g., diphenyl). The multiplet at 4.89–5.09 ppm is related to the proton signal at the tertiary carbon atoms (in position 4) in the TEMPO structure. The triplet at 2.10–2.40 ppm is related to the signal from protons near the carbonyl groups. The double doublet at 1.79–1.90 ppm is due to the signal from the protons of the piperidine ring at positions 3 and 5. The multiplet at 1.18–1.26 ppm is due to the signal from the middle hydrogen atoms in the sebacic acid chain. The pronounced doublet with peaks at 1.14 and 1.17 ppm is due to the signal from the protons in the CH_3_-groups at positions 2 and 6 of the piperidine ring. Finally, hydroxyl protons show a broadened signal centered at 4.28 ppm. The proton chemical shift values are generally consistent with the results given in [16].

The ^13^C spectra are normalized to the peak at 77.16 ppm of CDCl_3_ (Figure 1b). The highest chemical shift is observed for carbon atoms in the carboxyl groups (173.5 ppm), followed by a series of peaks (151.3, 129.3, 128.4, 119.6, and 112.3 ppm) that correspond to signals from carbon atoms in phenylhydrazine molecules and its oxidation products. The next peaks correspond to carbon atoms of the piperidine ring in positions 4 (66.2 ppm), 2 and 6 (60.7 ppm), and 5 and 3 (40.6 ppm). The next series of peaks are attributed to carbon atoms in the sebacic acid chain in order of distance from the carboxyl group (34.6, 31.3, 25.0, and 20.9 ppm). The signals from the carbon atoms of the methyl groups of the piperidine ring occupy an intermediate position among them with a chemical shift of 29.2 ppm.

### 3.2. Delay in the Onset of Cross-Linking of Polymer Compositions

Applying measuring mixers is a powerful experimental method for studying the kinetics of crosslinking and curing processes [23,24]. Figure 2a shows plastograms of dynamic peroxide crosslinking of LLDPE/POE blends with different bis-TEMPO content, representing the dependence of the torque on the mixer shaft on time, while the temperature in the mixing chamber is registered. During the initial stages of mixing (up to 1 min), plastograms show fluctuations due to non-uniform temperature and composition of the polymer melt. Subsequently, the melt parameters stabilize, and the crosslinking process begins. The rise in torque is associated with an increase in the viscosity of the polymer blend during the crosslinking process, which is due to an increase in the average molecular weight of the crosslinked product.

Bis-TEMPO-free composition (XPE_0) shows a crosslinking onset of 92 s after the loading of polymers and additives, with the torque increasing sharply, reaching an approximately constant value of about 41.1 N·m. The temperature of the polymer blend also changes during the crosslinking process. The initial temperature of the mixer chamber was 180 °C, which decreases when the polymer granules are loaded to 140 °C due to the consumption of the latent heat of fusion of the polymers, then returns to 170 °C, and further reaches 211 °C during the crosslinking process. For the XPE_0.11 composition, the onset of crosslinking is observed after 108 s; i.e., there is a 16 s delay compared to the XPE_0 composition. In this case, changing patterns of the torque and temperature vs. time are close to that of the XPE_0 composition, although a lower torque of 37.4 N·m and a lower temperature of 206 °C are achieved. The XPE_0.22 composition demonstrates the onset of crosslinking after 118 s, thus giving a delay of 26 s while reaching a torque of 38.8 N·m and a temperature of 207 °C. An increase in the bis-TEMPO content in the polymer mixture to 0.27 phr (XPE_0.27 composition) leads to an even greater delay in the onset of crosslinking to 36 s, with a final torque of 38.6 N·m and a temperature of 208 °C. Interestingly, the XPE_0.33 and XPE_0.44 compositions demonstrate a breakdown of the crosslinking, showing only a short-term increase in torque and reaching a plateau at values of 16.4 and 14.5 N·m, respectively. The final temperature after 15 min mixing of XPE_0.33 and XPE_0.44 compositions is also lowered down to 193 and 191 °C, respectively. The delay in the onset of crosslinking can be estimated at approximately 28 and 52 s for the XPE_0.33 and XPE_0.44 compositions, respectively. The crosslinking delay onset times for each of the compositions are also listed in Table 2.

Since the viscosity of the polymer blend depends on temperature, and the the temperature varies within a certain range during the crosslinking process, the obtained plastograms should be calculated in the isothermal mode for a more correct display of the crosslinking kinetics. Figure 2b shows corrected isothermal torque curves (for 190 °C), from which it is evident that the composition XPE_0 is characterized by a continuous increase in viscosity during the crosslinking process, while the compositions XPE_0.11–XPE_0.27 have small peaks of increasd viscosity at the beginning of crosslinking, followed by a decrease in viscosity and then a smooth increase with gradual steadying.

### 3.3. Dynamic Mechanical Analysis

Figure 3a shows the temperature dependence of the storage modulus ($E^{\prime }$) of the compositions in the temperature range of 25–90 °C. As can be seen, the $E^{\prime }$ values near RT correlate with the degree of crosslinking of the compositions, taking maximum values (53 MPa) for non-crosslinked compositions XPE_0.33 and XPE_0.44; the minimum value (30 MPa) falls on the crosslinked composition XPE_0 without bis-TEMPO. It is interesting to note that the addition of bis-TEMPO increases the $E^{\prime }$ of the crosslinked samples, which is evident from the RT values for compositions XPE_2 (32.2 MPa), XPE_3 (33.3 MPa), and XPE_4 (38.5 MPa). With an increase in temperature, the storage modulus expectedly decreases for all samples. However, the noted trend remains: for example, at a temperature of 70 °C, the lowest storage modulus is demonstrated by the crosslinked composition without bis-TEMPO (XPE_0), while the XPE_0.11 and XPE_0.22 compositions have a higher storage modulus by 6% and 9%. The increase in storage modulus reflects a rise in crosslinking density due to the auxiliary role of bis-TEMPO molecules in crosslinking. The relationship between crosslinking density ($\nu_{s}$) and storage modulus is quantified by the formula $\nu_{s}=E^{\prime }/RT$, where $E^{\prime }$ is the storage modulus in the rubbery plateau, *R* is the universal gas constant, and *T* is absolute temperature [25].

Figure 3b demonstrates the temperature dependence of the mechanical loss tangent in the same temperature range. The obtained curves for non-crosslinked compositions are S-shaped, with a maximum apparently located at a temperature above 90 °C. The curves for crosslinked compositions are characterized by lower values of the tangent and a bell-shaped form. The maximum for compositions XPE_0–XPE_0.22 is observed at 57–58 °C, and for composition XPE_0.27 at 61 °C. The maxima on the obtained curves are associated with α-relaxation processes in polyolefins, caused by the movement of large fragments of molecules in crystalline regions. A decrease in the α-transition temperature correlates with a decrease in the crystallinity of the polymer.

### 3.4. Anti-Oxidative Effect of bis-TEMPO

The study of oxidative stability was carried out by determining the thermal effect of oxidation of polymer samples by oxygen under certain conditions using the DSC method. It is evident from Figure 4a that the sample crosslinked without the addition of bis-TEMPO (XPE_0) demonstrates a sharp thermal effect associated with oxidation after just 1 min of exposure at 200 °C in an oxygen atmosphere. The addition of bis-TEMPO even in small quantities (0.11–0.27 phr) slows down the rate of oxidation of polymer samples, increasing their OIT time by 1.7–2.2 times depending on the bis-TEMPO content. The program-calculated values of the OIT for the studied samples are presented in Table 2. However, it should be noted that a significant difference in the effect of bis-TEMPO is found in the oxidative stability of crosslinked (XPE_0–XPE_0.27) and non-crosslinked (XPE_0.33 and XPE_0.44) compositions. In the first case, the effect consists of a moderate increase in OIT, while in the second case, the effect of increasing the oxidative stability is so pronounced that the oxidation of polymer samples slows down extremely. For clarity, we can compare the XPE_0.27 and XPE_0.33 compositions, which contain slightly different amounts of bis-TEMPO, while a dramatic difference in the OIT of 4.8 min and more than 40 min is observed. The studied antioxidant effect may be associated with both the presence of free-radical bis-TEMPO molecules and the product of their reaction with alkyl radicals (N-alkoxyamines), which are also effective antioxidants.

The oxidation onset temperature (OOT) of XPE_0–XPE_0.44 compositions (Figure 4b) also correlates with its OIT values discussed above. The addition of bis-TEMPO to polymer blends moderately increases the OOT for crosslinked samples, while for non-crosslinked samples, a dramatic increase in OOT is also observed (228 °C vs. 255 °C for compositions XPE_0.27 and XPE_0.33, respectively, which is a difference of 27 °C). Interestingly, non-crosslinked compositions exhibit a distinct second peak of the thermal effect at a higher temperature of about 370 °C, implying a two-stage oxidation process. The OOT times calculated by the software for the studied samples are also presented in Table 2.

## 4. Discussion

Crosslinking of polymer blends based on LLDPE and POE performed in this study is initiated by thermally activated decomposition of organic peroxide. The thermal decomposition paths of DBPH have been investigated in several works [26,27]. The first stage may involve the cleavage of the DBPH molecule by the −O−O− bonds with the sequential formation of two radicals and one biradical (Figure 5). Further reactions may include the reaction of a tert-butoxy radical with the polymer chain, resulting in abstraction of hydrogen from the carbon of the alkyl chain and formation of a tert-butyl alcohol molecule, or further fragmentation of the tert-butoxy radical with the formation of an acetone molecule and a CH_3_^•^ radical, which then also participates in the abstraction of hydrogen from the polyolefin chain with the formation of a CH_4_ molecule and a macroradical. Biradical can also undergo similar reactions. In total, DBPH decomposition leads to the formation of four radicals per molecule and a number of low-molecular by-products.

Crosslinking of polymer chains is caused by different types of processes: direct termination of macroradicals (Figure 6), which is a less probable process, and crosslinking by grafting of TAIC molecules, which is a more probable process and leads to a more branched polymer network (Figure 7). The possible participation of bis-TEMPO molecules in crosslinking is shown in Figure 8 and consists of recombination of the biradical bis-TEMPO molecule with two macroradicals. Side processes that do not lead to crosslinking, such as the capture of methyl radicals with the formation of bis(1-methoxy-TEMPO) sebacate, as well as one-sided grafting of bis-TEMPO to the polymer chain with its termination by a methyl radical, are also possible.

Following [13], the delay in the onset of crosslinking using TEMPO derivatives is determined by the parameter “trapping ratio”, which is the ratio of the number of stable TEMPO radicals to the number of radicals generated by peroxide decomposition. The DBPH concentration in the compositions corresponding to 0.5 phr is 17.2 μmol per gram of polymer blend. In this case, the concentration of bis-TEMPO varied in the range of 0.11–0.44 phr, or 2.16–8.64 μmol/g. To calculate the trapping ratio (ζ), we also take into account that the bis-TEMPO molecule is a biradical.(1)$$\zeta =\frac{\left[\mathrm{bis}-\mathrm{TEMPO}\right]}{2[\mathrm{DBPH}]}$$

Considering also that the delay in the onset of crosslinking ($t_{\mathrm{ind}}$) is determined by the rate of peroxide decomposition, which is described by a first-order kinetic equation, we can obtain the relationship relating the delay time and the trapping ratio:(2)$$t_{\mathrm{ind}}=-\frac{1}{k_{d}}\ln\left(1-\zeta \right)$$
where $k_{d}$ is the first-order rate constant for initiator homolysis at the reaction temperature.

Plotting the obtained values for the delay in the onset of crosslinking in the coordinates of Equation (Equation 2) yields a linear dependence for concentrations within 0–0.27 phr bis-TEMPO, which confirms the above-described delay mechanism (Figure 9). The deviation of two points at 0.33 and 0.44 phr from this linear dependence is explained by the reduced temperature of the polymer mixture when introducing these concentrations (about 191–193 versus 207–211 °C for compositions with 0–0.27 phr bis-TEMPO, see Figure 2). Thus, Equation (Equation 2) is applicable for estimating the delay time of crosslinking of polymer blends under dynamic, nonequilibrium conditions implemented in Brabender-type mixers at constant temperature. It is interesting to note that in the case under discussion, the introduction of more than 0.33 phr bis-TEMPO (ζ = 0.125) leads to a complete breakdown of crosslinking, while in [7,16], for ζ = 0.25, it was possible to obtain crosslinked samples, albeit with a reduced crosslinking density.

The ultimate goal of the use of scorching retardants (in particular, TEMPO derivatives) is to improve the homogeneity and mechanical properties of the resulting crosslinked polymers. In Section 3.3, we have already demonstrated that for compositions crosslinked with the addition of bis-TEMPO, there is an increase in the storage modulus compared to compositions crosslinked without bis-TEMPO. An additional method for studying the mechanical properties of polymers, closely related to their crosslinking density, is determining the ultimate tensile strength and elongation at break. Considering that this method usually gives some scattering of data, two series of crosslinked XPE_0 and XPE_0.22 compositions were made. The data obtained were processed statistically. Figure 10 shows that the addition of 0.22 phr bis-TEMPO on average increases the ultimate tensile strength of the crosslinked samples. The obtained values are 8.6 ± 0.8 MPa for compositions without bis-TEMPO and 9.9 ± 1.1 MPa for compositions with 0.22 phr bis-TEMPO. At the same time, the average elongation at break for both series of compositions has fairly close average values of 230 ± 45% and 251 ± 57%, respectively. Thus, it was evident that the mechanical properties of the crosslinked samples were improved by the addition of bis-TEMPO; however, no impact on the homogeneity of the samples due to bis-TEMPO was observed.

Taking into account an increase in the gel fraction content (Table 2) in XPE compositions, it can be suggested that the addition of bis-TEMPO to polymer blends leads to an increase in the crosslinking density of polyolefine compositions. The most significant increase in the gel fraction from 72 to 80% is observed when using a bis-TEMPO concentration of 0.22 phr, which apparently demonstrates an optimum between the rates of radical quenching and polymer crosslinking.

However, the reason for such an increase in the gel fraction remains an open question. This could be attributed to two distinct factors: (1) the auxiliary participation of bis-TEMPO in the crosslinking reactions; and (2) an increase in the crosslinking delay time, which promotes a more complete and uniform distribution of the peroxide in the polymer before the onset of crosslinking. To clarify this issue, we studied the crosslinking of similar polyolefin composites under the same conditions using a monoradical derivative of TEMPO, 4-benzoyloxy-TEMPO (BzO-TEMPO). It was experimentally established that BzO-TEMPO at a concentration of 0.11 phr leads to approximately the same delay in the onset of the crosslinking process as when using 0.27 phr bis-TEMPO (Figure 11). However, the gel fraction in the samples crosslinked using BzO-TEMPO is significantly reduced and averages 62%, while for a composition crosslinked with the addition of 0.27 phr bis-TEMPO with a same delay in onset, the gel content is 74%. Thus, the increase in the delay in the onset of crosslinking is not the main factor influencing the crosslinking density, but rather it is more likely to be the participation of bis-TEMPO molecules in the crosslinking processes, which, in turn, affects the increase in the gel fraction.

In this article, we have established that, while being an effective scorch retardant that slows down the peroxide crosslinking process while increasing its density, bis-TEMPO also imparts an antioxidant effect to crosslinkable composites, which was quantitatively characterized in Section 3.4. Although the antioxidant effect of nitroxides is widely known in the literature [28,29], its application to crosslinkable composites is less frequently discussed; for monoradical TEMPO derivatives, the simultaneous effect on crosslinking density and antioxidant stability was investigated by Twigg C. et al. [30]. Presumably, the antioxidant activity of bis-TEMPO in our case is due to the products of its reaction with radicals and macroradicals that form an N-alkoxyamine bond. Interestingly, antioxidants based on hindered amine stabilizers (HAS), often used in industry, are not capable of effectively delaying the onset of crosslinking. Figure 11 also shows a crosslinking plastogram for a sample based on the XPE_0 composition containing 0.22 phr of Tinuvin 770 (bis(2,2,6,6-tetramethyl-4-piperidyl) sebacate), which is the closest amine analogue of the bis-TEMPO used. It can be seen that the addition of Tinuvin 770 even slightly accelerates the crosslinking process, although a significant drop in the final torque is observed, indicating a decrease in the crosslinking density. Thus, bis-TEMPO is a unique bifunctional additive for scorch suppressing, delaying the crosslinking process, while increasing the crosslinking density and enhancing the oxidative stability of crosslinked polyolefin compositions.

## 5. Conclusions

In this work, it is demonstrated that during peroxide-initiated crosslinking of polymer composites based on LDPE and POE in a Brabender measuring mixer the use of bis-TEMPO as a scorching inhibitor in the concentration range up to 0.11–0.27 phr leads to a predictable delay in the onset of crosslinking, while concentrations above 0.33 phr completely inhibit the crosslinking process, leading to non-crosslinked composites. For polymer blends crosslinked with the addition of bis-TEMPO, an increase in the storage modulus, gel fraction and tensile strength is observed compared to similar polymer blends crosslinked without the addition of bis-TEMPO, which is a consequence of the increase in the crosslinking density. Moreover, the addition of bis-TEMPO to the polymer blends also improves their oxidative stability, although it turns out to be especially pronounced for non-crosslinked composites.

## Figures and Tables

**Figure 1 polymers-17-03325-f001:**
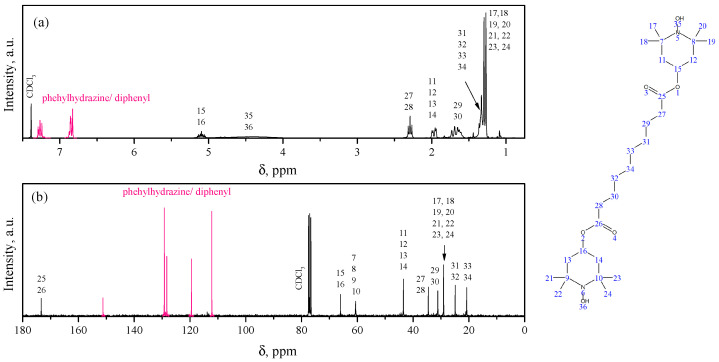
^1^H (**a**) and ^13^C (**b**) NMR spectra of the product of reduction of synthesized bis-TEMPO with phenylhydrazine in CDCl_3_.

**Figure 2 polymers-17-03325-f002:**
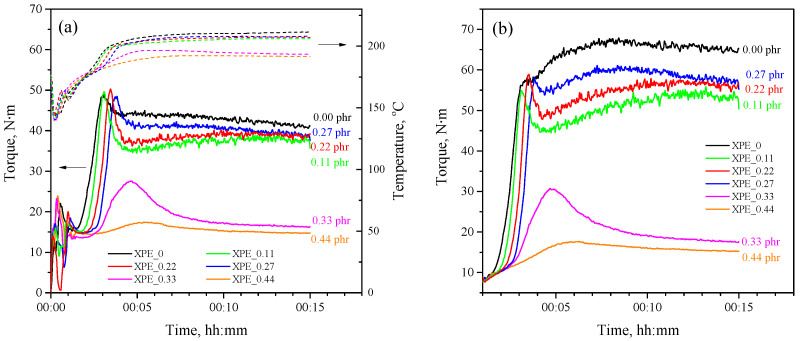
(**a**) Experimental plastograms of the dynamically crosslinking XPE_0–XPE_0.44 compositions (left axis) and corresponding temperature profiles during crosslinking process (right axis); (**b**) calculated isothermal plastograms for the same compositions.

**Figure 3 polymers-17-03325-f003:**
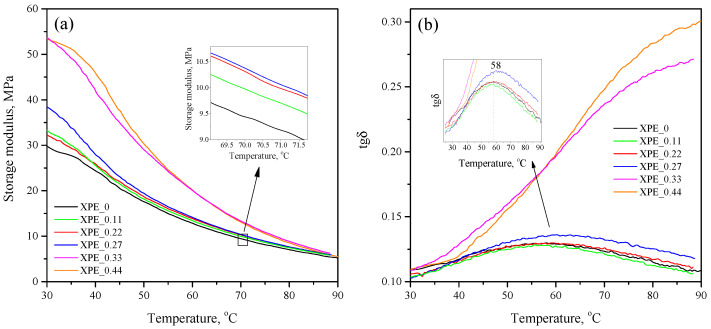
(**a**) Temperature dependence of the storage modulus (real part of the complex modulus) of the of the XPE_0–XPE_0.44 compositions; (**b**) temperature dependence of the mechanical loss tangent of the of the XPE_0–XPE_0.44 compositions.

**Figure 4 polymers-17-03325-f004:**
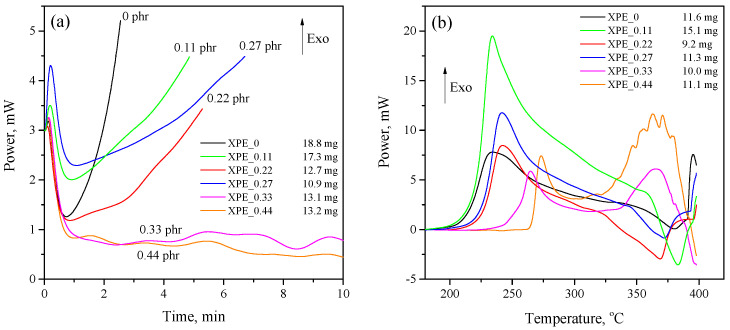
(**a**) Determination of oxidation induction time (OIT) and (**b**) oxidation onset temperature (OOT) of the XPE_0–XPE_0.44 compositions.

**Figure 5 polymers-17-03325-f005:**
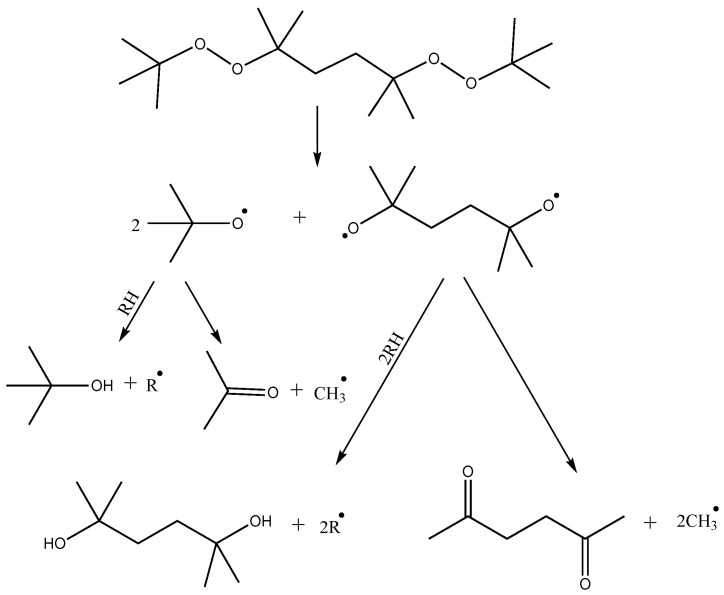
Reactions of thermal decomposition of DBPH, where R^•^ is macroradical.

**Figure 6 polymers-17-03325-f006:**
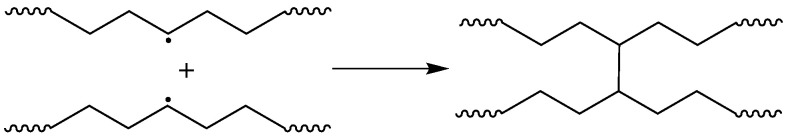
Crosslinking of polymer chains by termination of macroradicals.

**Figure 7 polymers-17-03325-f007:**
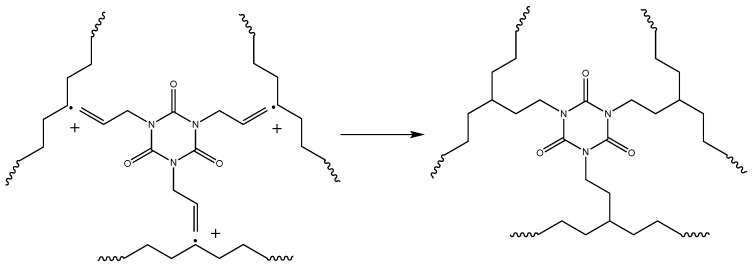
Crosslinking of polymer chains with the aid of crosslinking agent (TAIC).

**Figure 8 polymers-17-03325-f008:**
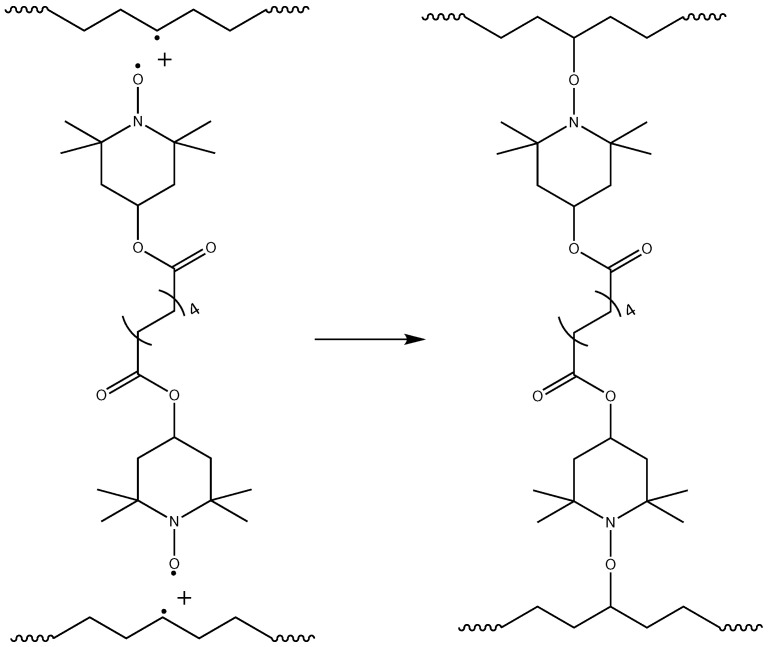
Crosslinking polymer chains through the recombination of of bis-TEMPO molecules with macroradicals.

**Figure 9 polymers-17-03325-f009:**
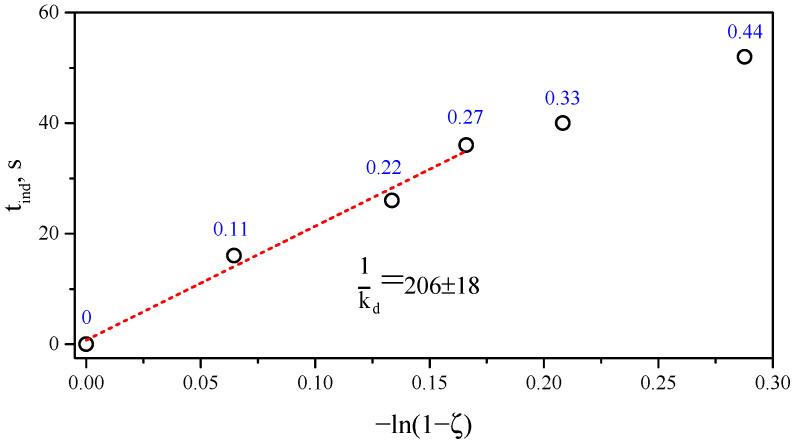
Dependence of induction time on the trapping ratio (ζ) in the coordinates of Equation (Equation 2). The bis-TEMPO content is marked in blue text near the points.

**Figure 10 polymers-17-03325-f010:**
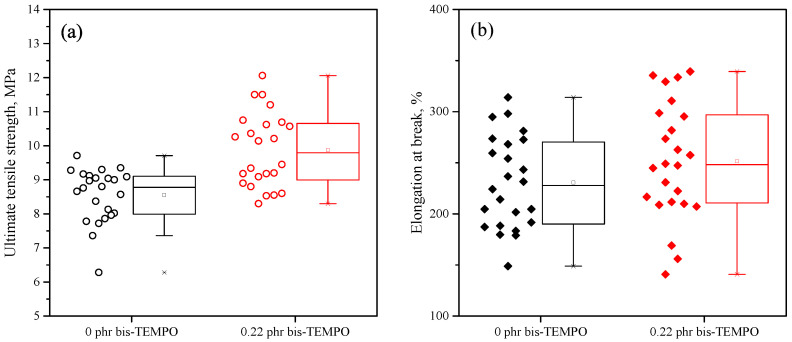
Statistical diagrams showing the ultimate tensile strength (**a**) and elongation at break (**b**) of sets of samples, crosslinked without the addition of bis-TEMPO and with 0.22 phr of bis-TEMPO.

**Figure 11 polymers-17-03325-f011:**
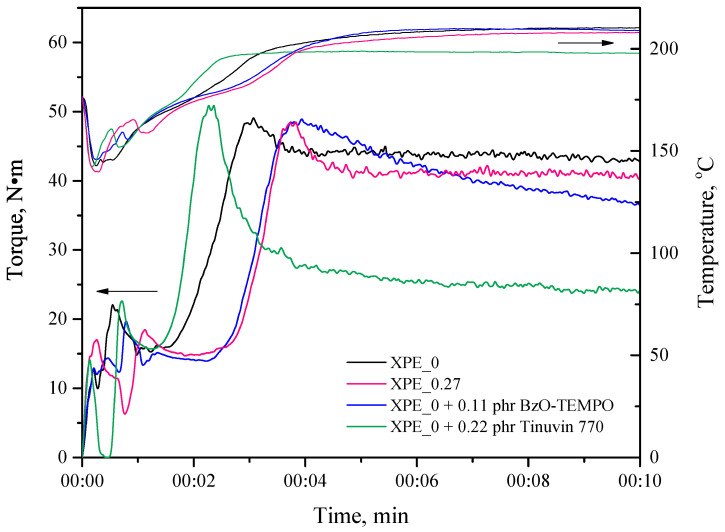
Experimental plastograms (left axis) of crosslinking of XPE compositions without the addition of scorch retardant (XPE_0) and with the addition of 0.27 phr bis-TEMPO (XPE_0.27), 0.11 phr of BzO-TEMPO, and 0.22 phr of Tinuvin 770 and corresponding temperature profiles during crosslinking process (right axis).

**Table 1 polymers-17-03325-t001:** Cross-linked compositions given in *phr*.

Components	XPE_0	XPE_0.11	XPE_0.22	XPE_0.27	XPE_0.33	XPE_0.44
LLDPE HP3518CN	30	30	30	30	30	30
POE LC670	70	70	70	70	70	70
Sonox^®^ 168	0.4	0.4	0.4	0.4	0.4	0.4
Irganox^®^ 1010 FF	0.2	0.2	0.2	0.2	0.2	0.2
TAIC	1	1	1	1	1	1
Trigonox^®^ 101	0.5	0.5	0.5	0.5	0.5	0.5
bis-TEMPO	0	0.11	0.22	0.27	0.33	0.44

**Table 2 polymers-17-03325-t002:** Experimental characteristics of the studied crosslinked compositions.

Composition	DOC ^1^, s	GC ^2^, %	OIT ^3^, min	OOT ^4^, °C	UTS ^5^, MPa	EB ^6^, %
XPE_0	0	72	2.2	217	8.3 ± 1.4	275 ± 35
XPE_0.11	16	70	3.8	221	9.5 ± 0.2	263 ± 9
XPE_0.22	26	80	3.7	227	11.5 ± 0.4	317 ± 23
XPE_0.27	36	74	4.8	228	9.8 ± 0.8	276 ± 30
XPE_0.33	40	0	>40	255	11.4 ± 0.6	664 ± 23
XPE_0.44	52	0	>40	267	11.9 ± 1.5	744 ± 50

^1^ Delay in the onset of crosslinking; ^2^ gel content; ^3^ oxidation induction time; ^4^ oxidation onset temperature; ^5^ ultimate tensile strength; ^6^ elongation at break.

## Data Availability

The original contributions presented in this study are included in the article. Further inquiries can be directed to the corresponding author.

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
