# Peer review of "Effect of bis-TEMPO Sebacate on Mechanical Properties and Oxidative Resistance of Peroxide-Crosslinked Polyolefin Compositions"

_polymers, 2025, doi:10.3390/polym17243325_

Round 1
Reviewer 1 Report
Comments and Suggestions for Authors
The paper by A. Chizhov et al. describes the effect of bis-Tempo-sebacate on the properties of cross-linked polyethylene. Although the experiments are well described and the conclusions are reasonable there is a substantial gap with regard to state-of-the art considerations. The use of the sebacate for scorch and premature cross-linking resistance is well-known from the patent literature (e.g. WO2002028946, US20040195550) and should be discussed. Moreover the sebacate is not novel and preparation and analytical results of the sebacate are described in many previous papers, the analytical results should be compared. The antioxidative effect of nitroxyl radicals is known for many years, therefore previous literature on the topic should be included and discussed and the novelty elaborated. Sources of the materials are missing. Standard deviations of the experimental results e.g. in table 2 should be added or discussed within the paper.
Author Response
Reviever's Response: The paper by A. Chizhov et al. describes the effect of bis-Tempo-sebacate on the properties of cross-linked polyethylene. Although the experiments are well described and the conclusions are reasonable there is a substantial gap with regard to state-of-the art considerations. The use of the sebacate for scorch and premature cross-linking resistance is well-known from the patent literature (e.g. WO2002028946, US20040195550) and should be discussed. Moreover the sebacate is not novel and preparation and analytical results of the sebacate are described in many previous papers, the analytical results should be compared. The antioxidative effect of nitroxyl radicals is known for many years, therefore previous literature on the topic should be included and discussed and the novelty elaborated. Sources of the materials are missing. Standard deviations of the experimental results e.g. in table 2 should be added or discussed within the paper.
Answer: Dear Reviewer, thank you for your attention to our article and valuable comments, which significantly helped improve the article. We agree that the use of bis-TEMPO to prevent scorching is not new and emphasized this in our introduction. The antioxidant effect of nitroxides is also known; however, nitroxides as antioxidants are rarely used in industry, and their amine analogs (hindered amines stabilizers, HAS, such as Tinuvin 770) are more commonly used. However, HAS do not delay the onset of crosslinking since they do not react with alkyl radicals directly. Therefore, we believe it was important to investigate the combined effect of bis-TEMPO on both crosslinking delay and oxidation resistance. To best of our knowledge, published studies examine either only the scorch retarding properties of bis-TEMPO or only the antioxidant properties of nitroxides; however, in real-world use, it's worth considering how bis-TEMPO affects both of these parameters. This is the novelty of our work too. Below are point-by-point responses to your comments.
Comment 1: The use of the sebacate for scorch and premature cross-linking resistance is well-known from the patent literature (e.g. WO2002028946, US20040195550) and should be discussed.
Response 1: The following addition have been made to the manuscript based on your comment:
“Some patents, such as [17], also have suggested the application of bis-TEMPO for its properties as a scorch retardant.”, (lines 79-81)
Comment 2: Moreover the sebacate is not novel and preparation and analytical results of the sebacate are described in many previous papers, the analytical results should be compared.
Response 2: Comparison of NMR data of synthesized bis-TEMPO with literature sources was added:
The proton chemical shift values are generally consistent with the results given in [16]. (lines 193-194)
Comment 3: The antioxidative effect of nitroxyl radicals is known for many years, therefore previous literature on the topic should be included and discussed and the novelty elaborated.
Response 3: We added a paragraph discussing the relationship between the scorch retarding and antioxidant properties of bis-TEMPO and added a comparison with its amine analogue Tinuvin 770 to Figure 11:
"In this article, we have established that, while being an effective scorch retardant that slows down the peroxide crosslinking process while increasing its density, bis-TEMPO also imparts an antioxidant effect to crosslinkable composites, which was quantitatively characterized in Section 3.4. Although the antioxidant effect of nitroxides is widely known in the literature [28,29], its application to crosslinkable composites is less frequently discussed; for monoradical TEMPO derivatives, the simultaneous effect on crosslinking density and antioxidant stability was investigated by Twigg C. et al [30]. Presumably, the antioxidant activity of bis-TEMPO in our case is due to the products of its reaction with radicals and macroradicals that form an N-alkoxyamine bond. Interestingly, antioxidants based on hindered amine stabilizers (HAS), often used in industry, are not capable of effectively delaying the onset of crosslinking. The Figure 11 also shows a cross-linking plastogram for a composition containing 0.22 phr of Tinuvin 770 (Bis(2,2,6,6-tetramethyl-4-piperidyl) sebacate), which is the closest amine analogue of the bis-TEMPO used. It can be seen that the addition of Tinuvin even slightly accelerates the cross-linking process, although a significant drop in the final torque is observed, indicating a decrease in cross-linking density. Thus, bis-TEMPO is a unique bifunctional additive for scorch suppressing, delaying the crosslinking process, while increasing the crosslinking density and enhancing the oxidative stability of crosslinked polyolefin compositions." (lines 381-398)
Comment 4: Sources of the materials are missing.
Response 4: Sources of materials were added in Section 2.1. (lines 99-102)
Comment 5: Standard deviations of the experimental results e.g. in table 2 should be added or discussed within the paper.
Response 5: Standard deviations of ultimate tensile stretch and elongation at break values were added in Table 2. Temperature measurement error in OIT and OOT studies was added to Section 2.4 (lines 168-169) . Gel fraction and crosslinking delay were not averaged and are characterized for individual samples.
Reviewer 2 Report
Comments and Suggestions for Authors
Comment 1: The abstract section needs to be revised. What is the hypothesis of the study and why is it a priority? It would be beneficial to add sentences that clearly state the aims and justification of the study.
” The laboratory synthesis route and NMR characterization of 14
bis-TEMPO are also given in detail.”
A detailed description of the laboratory synthesis route and NMR characterization of bis-TEMPO cannot be the final sentence for the abstract; end with a sentence that provides a perspective on the contribution of the study.
Comment 2, Page 1, line 29: “in combination with with” Page 3, line 114-118 “Than” Check for spelling errors
Comment 3, Page 2, line 42: Why delayed crosslinking leads to a decrease in crosslinking density requires detailed discussion. Isn't it noted that in some hydrogel systems, delayed crosslinking results in more homogeneous crosslinking?
Comment 4, Page 2, line 48: How can the residues of the α-methylstyrene moiety grafted onto the polymer chain be expected to activate the crosslinking process, should they not undergo transformation in the pendant state, and how can they compensate for the loss of effective crosslinking density?
Comment 5, Page 3, line 83-90: Revise the last section of the introduction by adding an explanation of why it is necessary to study the effect of bis-TEMPO at part by weight concentration on the delay in the onset of dynamic crosslinking in linear low-density polyethylene (LLDPE) and polyolefin elastomer (POE) based polymer blends.
Comment 6, Page 5, line 207: While the initial temperature of the mixer chamber is 180°C, this temperature drops when the polymer granules are loaded to 140°C, then increases again at the beginning of crosslinking. Please explain how this was determined in the text.
Comment 7, Page 6, line 216: The increase in torque is attributed to the increase in viscosity of the polymer blend during the crosslinking process, which is due to the two maximums observed in Figures 2 a and b. Explain by evaluating the polymerization kinetics.
Comment 8, Page 7, line 244: Please explain why the XPE_0.11 and XPE_0.22 combinations have 6% and 9% higher storage modulus in the text.
Comment 9, Page 7, line 272: Why is the greatest antioxidant effect associated with the free radical bis-TEMPO molecules preserved in the sample? Discuss further by supporting with literature.
Comment 10, Page 10, line 331: Provide a perspective to the reader by explaining what the combination of homogeneity and mechanical properties of the resulting cross-linked polymers is.
Comments on the Quality of English LanguageWhile some parts of the article are very well written, they need to be revised for simple grammatical errors such as:
Comment 2, Page 1, line 29: “in combination with with” Page 3, line 114-118 “Than” Check for spelling errors.
Author Response
Comment 1: The abstract section needs to be revised. What is the hypothesis of the study and why is it a priority? It would be beneficial to add sentences that clearly state the aims and justification of the study.
” The laboratory synthesis route and NMR characterization of bis-TEMPO are also given in detail.”
A detailed description of the laboratory synthesis route and NMR characterization of bis-TEMPO cannot be the final sentence for the abstract; end with a sentence that provides a perspective on the contribution of the study.
Response 1: Dear reviewer, thank you for your careful review of the abstract. We've attempted to rewrite last half of the abstract based on your comments:
"The aim of this work was to reveal the effect of bis(1-oxyl-2,2,6,6-tetramethylpiperidin-4-yl) sebacate (bis-TEMPO) in the concentration range of 0.11–0.44 phr on the delay in the onset of dynamic crosslinking of polyolefin composites initiated by peroxide, as well as the oxidative stability of the resulting crosslinked composites. The obtained data show that using bis-TEMPO at a concentration of less than 0.27 phr increases the crosslinking density of the polyolefin composite, with a crosslinking onset delay of up to 36 s achieved. At the same time, antioxidant effect of bis-TEMPO in crosslinked composites is moderate and leads to an increase in the OIT values by 1.7-2.8 times. The crosslinking onset delay time under dynamic conditions is well described by a first-order kinetic model at a constant temperature. The obtained data confirm the efficiency and predictability of bis-TEMPO as a scorching inhibitor for polyolefin composites." (lines 7-16)
Comment 2, Page 1, line 29: “in combination with with” Page 3, line 114-118 “Than” Check for spelling errors
Response 2: Dear reviewer, thank you for your careful reading of the text. Spell checking has been performed. The failed phrase has been replaced with the following text:
“Subsequently, the reaction mixture was maintained at 40 ◦C for a 2 h; following this, the heating was discontinued, and the reaction was allowed to proceed at room temperature (RT) for an additional 18 h. Each 1 L portion of the reaction mixture was then transferred into a 2 L separatory funnel and subjected to four washes with 0.5 L of a dilute HCl solution (10 mL HCl per 1 L of water). The organic phases were collected and dried over Na2SO4, and the solvent was removed using a rotary evaporator at 70oC, yielding a viscous dark red liquid. Finally, 300 ml of ethanol was added to this liquid, and the resulting solution was combined with 300 ml of water at RT.” (lines 119-127)
Comment 3, Page 2, line 42: Why delayed crosslinking leads to a decrease in crosslinking density requires detailed discussion. Isn't it noted that in some hydrogel systems, delayed crosslinking results in more homogeneous crosslinking?
Response 3: Thank you for your comment. The loss of cross-linking density when using TEMPO and its monoradical derivatives is due to the irreversible capture of alkyl radicals, which initiate the cross-linking process. Crosslinking of polymers is not a chain process, but a stoichiometric one, and directly depends on the concentration of alkyl radicals. The capture of a certain number of radicals is equivalent to a decrease in the peroxide concentration, so below a certain threshold concentration, cross-linking will break down. This issue is discussed at length by J. Scott Parent et al. in their articles, number of which we have cited in the manuscript ([13-16]).
Comment 4, Page 2, line 48: How can the residues of the α-methylstyrene moiety grafted onto the polymer chain be expected to activate the crosslinking process, should they not undergo transformation in the pendant state, and how can they compensate for the loss of effective crosslinking density?
Response 4: Thank you for your comment. We rely on the results of the cited article. The scorching inhibition process is a two-step process. In the first step, alpha-methylstyrene dimer (MSD) adds to macroradicals at the double bond, forming an intermediate adduct radical, which fragments to form a cumyl radical. The unsaturated fragment of the MSD molecule remains grafted to the polymer chain. The MSD residues grafted to the chain activate the cross-linking process because they are unsaturated. Than, the second step, these unsaturated groups react with macroradicals, producing a cross-linked polymer. The reaction scheme is given in the cited article.
The corresponding text in the manuscript has been replaced:
"To address this issue, an initial study suggested employing a methylstyrene dimer (MSD), which led to a delay in the onset of crosslinking as well as an enhancement in the crosslinking density. In the first step, alpha-methylstyrene dimer (MSD) adds to macroradicals at the double bond, forming an intermediate adduct radical, which fragments to form a cumyl radical, at the same time the unsaturated fragment of the MSD molecule remains grafted to the polymer chain. At the second step, these unsaturated MSD residues react with macroradicals, producing a cross-linked polymer." (lines 44-50)
Comment 5, Page 3, line 83-90: Revise the last section of the introduction by adding an explanation of why it is necessary to study the effect of bis-TEMPO at part by weight concentration on the delay in the onset of dynamic crosslinking in linear low-density polyethylene (LLDPE) and polyolefin elastomer (POE) based polymer blends.
Response 5: Thank you for your precise comment. LLDPE/POE mixtures with this 30/70 ratio are the base material with suitable mechanical properties for the production of unfilled insulation of electrical cables, and therefore were selected for study. Following text was added to the last section of Introduction:
"This composition was selected because it provides, after cross-linking, suitable mechanical properties for unfilled insulation of some types of electrical cables." (lines 88-90)
Comment 6, Page 5, line 207: While the initial temperature of the mixer chamber is 180°C, this temperature drops when the polymer granules are loaded to 140°C, then increases again at the beginning of crosslinking. Please explain how this was determined in the text.
Response 6: Thank you for your question. The initial chamber temperature was 180°C, but upon loading polymer granules and additives, it dropped to 140°C due to the consumption of the latent heat of fusion of the polymers, then returned to 170°C and more. Temperature measurements are taken simultaneously with torque measurements, so it can be established that heating the polymer mixture to 180°C correlates with the onset of crosslinking (ignoring the retarding effect of bis-TEMPO). Following text was added in manuscript:
"due to the consumption of the latent heat of fusion of the polymers, then returned to 170°C" (lines 219-220)
Comment 7, Page 6, line 216: The increase in torque is attributed to the increase in viscosity of the polymer blend during the crosslinking process, which is due to the two maximums observed in Figures 2 a and b. Explain by evaluating the polymerization kinetics.
Response 7: Thank you for your important question. If I correctly understood your question, it should be noted that the region of plastograms below 1 minute is generally uninformative due to effects associated with non-uniform melting and mixing of polymer grains. Probably, the effect you described can be explained as follows. In the first seconds after loading the granules, they gradually heat up and melt. At low temperatures (100-120°C), the melt has a high viscosity. However, as the melt heats up (but before crosslinking begins), its viscosity decreases. This causes the first peak to appear on the torque-time curve. Following text was added in the manuscript:
"During the initial stages of mixing (up to 1 min), plastograms show fluctuations due to non-uniform temperature and composition of the polymer melt. Subsequently, the melt parameters stabilize, and the cross-linking process begins." (lines 209-212)
Comment 8, Page 7, line 244: Please explain why the XPE_0.11 and XPE_0.22 combinations have 6% and 9% higher storage modulus in the text.
Response 8: Thank you for your question. We believe the higher modulus is due to the auxiliary role of bis-TEMPO molecules in cross-linking, resulting in an increase in the cross-linking density of these materials. Cross-linking density is known to be proportional to the storage modulus (at a constant temperature). The following text and link have been added to the manuscript:
The increase in storage modulus indicates an increase in cross-linking density due to the auxiliary role of bis-TEMPO molecules in cross-linking. The relationship between cross-linking density and storage modulus is determined by the formula νs = E′/RT, where E′ is the storage modulus in the rubbery plateau, R is the universal gas constant and T is absolute temperature [25]. (lines 257-261)
Comment 9, Page 7, line 272: Why is the greatest antioxidant effect associated with the free radical bis-TEMPO molecules preserved in the sample? Discuss further by supporting with literature.
Response 9: Indeed, we cannot claim that the greatest antioxidant effect is associated with the free-radical bis-TEMPO molecules preserved in the sample, as this requires additional EPR studies, which are not currently available for us. This statement will be removed from the text. The following text has been added in its place:
"The studied antioxidant effect may be associated with both the presence of free-radical bis-TEMPO molecules and the product of their reaction with alkyl radicals (N-alkoxyamines), which are also effective antioxidants." (lines 287-289)
Also, the discussion of the antioxidant functionality of bis-TEMPO is carried out in the text added by the demand of the 2nd Reviewer on lines 381-398.
Comment 10, Page 10, line 331: Provide a perspective to the reader by explaining what the combination of homogeneity and mechanical properties of the resulting cross-linked polymers is.
Response 10: Thank you for your precise question. Unfortunately, it is not possible to evaluate the effect of bis-TEMPO on the homogeneity of the cross-linked samples from our studies. It is possible that scorching would have required the addition of larger amounts of peroxide, higher heating of the polymer mixture, or poorer mixing. However, we clearly demonstrated that bis-TEMPO delays the onset of cross-linking, which strongly suggests that it will act as a scorch retardant under the appropriate conditions.
Even from statistical analyses of the ultimate tensile strength, we obtained approximately the same data scattering for samples with and without bis-TEMPO (Figure 10a). This indirectly indicates a similar degree of homogeneity between the samples. The following text has been added to the manuscript:
"Thus, it was evident that the mechanical properties of the cross-linked samples were improved by the addition of bis-TEMPO, however, no impact on the homogeneity of the samples due to bis-TEMPO was observed." (lines 356-359)
Round 2
Reviewer 1 Report
Comments and Suggestions for Authors
The revised version can be accepted in the present form, the comments were fully adressed.
Reviewer 2 Report
Comments and Suggestions for Authors
Comment 1: The abstract section needs to be revised. What is the hypothesis of the study and why is it a priority? It would be beneficial to add sentences that clearly state the aims and justification of the study.
” The laboratory synthesis route and NMR characterization of 14
bis-TEMPO are also given in detail.”
A detailed description of the laboratory synthesis route and NMR characterization of bis-TEMPO cannot be the final sentence for the abstract; end with a sentence that provides a perspective on the contribution of the study.
Response to Authors’s revision: The authors have attempted to rewrite last half of the abstract based on the suggested comments.
Comment 2, Page 1, line 29: “in combination with with” Page 3, line 114-118 “Than” Check for spelling errors
Response to Authors’s revision: The authors have checked the spelling and changed the incorrect expressions accordingly.
Comment 3, Page 2, line 42: Why delayed crosslinking leads to a decrease in crosslinking density requires detailed discussion. Isn't it noted that in some hydrogel systems, delayed crosslinking results in more homogeneous crosslinking?
Response to Authors’s revision: An explanation has been included for this comment.
Comment 4, Page 2, line 48: How can the residues of the α-methylstyrene moiety grafted onto the polymer chain be expected to activate the crosslinking process, should they not undergo transformation in the pendant state, and how can they compensate for the loss of effective crosslinking density?
Response to Authors’s revision: The answer was given by changing the relevant text in the manuscript.
Comment 5, Page 3, line 83-90: Revise the last section of the introduction by adding an explanation of why it is necessary to study the effect of bis-TEMPO at part by weight concentration on the delay in the onset of dynamic crosslinking in linear low-density polyethylene (LLDPE) and polyolefin elastomer (POE) based polymer blends.
Response to Authors’s revision: A revision was made by adding a text to the last section of the introduction.
Comment 6, Page 5, line 207: While the initial temperature of the mixer chamber is 180°C, this temperature drops when the polymer granules are loaded to 140°C, then increases again at the beginning of crosslinking. Please explain how this was determined in the text.
Response to Authors’s revision: An explanation was added in the revised manuscript.
Comment 7, Page 6, line 216: The increase in torque is attributed to the increase in viscosity of the polymer blend during the crosslinking process, which is due to the two maximums observed in Figures 2 a and b. Explain by evaluating the polymerization kinetics.
Response to Authors’s revision: An explanation has been added to discuss in terms of the polymerization kinetics.
Comment 8, Page 7, line 244: Please explain why the XPE_0.11 and XPE_0.22 combinations have 6% and 9% higher storage modulus in the text.
Response to Authors’s revision: The revision was completed by adding additional text and links to the article.
Comment 9, Page 7, line 272: Why is the greatest antioxidant effect associated with the free radical bis-TEMPO molecules preserved in the sample? Discuss further by supporting with literature.
Response to Authors’s revision: Based on the comment, the relevant statement was removed from the text and a revised text was added instead.
Comment 10, Page 10, line 331: Provide a perspective to the reader by explaining what the combination of homogeneity and mechanical properties of the resulting cross-linked polymers is.
Response to Authors’s revision: An explanation has been added to the manuscript to provide further conclusion based on the comment.